# Biomarkers as Beacons: Illuminating Sepsis-Associated Hepato-Renal Injury

**DOI:** 10.3390/ijms26104825

**Published:** 2025-05-18

**Authors:** Maria-Antoanela Pasare, Cristian Sorin Prepeliuc, Maria Gabriela Grigoriu, Ionela-Larisa Miftode, Egidia Gabriela Miftode

**Affiliations:** 1Doctoral School, “Grigore T. Popa” University of Medicine and Pharmacy, 700115 Iasi, Romania; antoanela.pasare@gmail.com (M.-A.P.); cristiprep94@gmail.com (C.S.P.); 2“Sf Parascheva” Clinical Hospital of Infectious Diseases, 700116 Iasi, Romania; grigoriumaria.mg@gmail.com (M.G.G.); emiftode@yahoo.co.uk (E.G.M.); 3Emergency Hospital “Mavromati”, 710221 Botosani, Romania; 4Department of Infectious Diseases, Faculty of Medicine, “Grigore T. Popa” University of Medicine and Pharmacy, 700115 Iasi, Romania

**Keywords:** sepsis, acute kidney injury, acute liver failure, IL-18, hepcidin, ferritin, IL-27

## Abstract

Sepsis, defined as a dysregulated host response to infection, is one of the leading causes of mortality worldwide. It unleashes in the organism a cascade of molecules, cytokines, and proteins which leads to an inflammatory storm. If this response to infection is uncontrolled, any organ is susceptible to damage. Acute kidney injury (AKI) is one of the most frequent organ dysfunctions in septic patients, and while it can be reversible, its presence leads to a higher burden of morbidity and mortality. While serum creatinine is essential in evaluating kidney function, the pathophysiology of AKI is not completely elucidated, and a plethora of novel biomarkers have been studied in the hope of an early diagnosis and fast treatment. While the liver is not as affected by sepsis, it plays an important role as a guardian by providing acute-phase proteins, activating neutrophils, and controlling iron balance. Acute liver failure (ALF) could impair the organism’s capacity to contain and eliminate pathogens. Some molecules have been associated with either AKI or ALF, although biomarkers specific for organ dysfunction are difficult to validate. The aim of this review is to understand the role of several molecules in the pathophysiology of sepsis and their clinical ability for diagnosing or predicting sepsis-induced hepato-renal dysfunction.

## 1. Introduction

To our best understanding today, sepsis represents a life-threatening organ dysfunction that is caused by a dysregulated host response to infection [1]. The task force that was appointed to define sepsis recommends using the SOFA (sequential organ failure assessment) scoring system to evaluate organ dysfunction in a patient with a suspected or confirmed infection [2].

Sepsis is one of the major contributors to the global health burden, alongside stroke, heart disease, and kidney disease [3,4]. Recent data indicate that sepsis incidence rates vary between 500 and 1500 cases per 100,000 people. Mortality rates are lower in high-income countries, between 15 and 25%, and can reach up to 40% in low- and middle-income countries [5]. Further epidemiological studies will elucidate sepsis’s incidence and health impact through the coronavirus disease (COVID-19) pandemic years, as some have categorized sepsis while referring to viral sepsis and some as a bacterial infection associated with COVID-19 [6].

On a cellular level, sepsis starts with the recognition of pathogen-derived molecular patterns (PAMPs) and damage-associated molecular patterns (DAMPs), which bind to receptors of antigen-presenting cells. Immune cells release a plethora of cytokines, both pro-inflammatory: such as interleukin (IL)-1, IL-6, IL-12, IL-18, and tumor necrosis factor alpha (TNF-α), and anti-inflammatory: IL-10, IL-27, and transforming growth factor beta 1 (TGFβ1). Inflammation leads to the release of acute-phase proteins by the liver, which in turn determines immune cell recruitment, activation of the complement system, the coagulation pathway, and iron regulation. Following the response from the innate immune system, the adaptive immune system comes into play as the organism tries to resolve the disease [7,8]. Part of the pathophysiology of sepsis and the interplay of various molecules is synthesized in Figure 1.

The delicate balance of inflammation and anti-inflammation is disrupted in sepsis, and while both pathways are upregulated, the resulting inflammation triggers progressive tissue damage and multi-organ dysfunction [7]. The kidneys and the liver are highly susceptible to microcirculatory dysfunction, and sepsis complicated with either hepatic or renal injury leads to high mortality rates, from 38% to almost 50% [8,9]. Hepato-renal dysfunction can exacerbate injury to other organs and can lead to an influx of toxins into the central nervous system. Additionally, coagulopathy can produce cerebral ischemia or hemorrhage [10]. Therefore, it is imperative to recognize sepsis-associated acute kidney injury (SA-AKI) and sepsis-associated liver injury (SALI) in a timely manner. Classical methods to detect kidney injury, such as creatinine and urine output, are not sufficiently sensitive, and their changes do not occur early enough to detect declines in renal function [11]. Liver biomarkers, such as bilirubin, aminotransferases, and alkaline phosphatase, are not satisfactory in their ability to reflect liver dysfunction, to differentiate acute from chronic injury, and to predict mortality [12].

As a result, in the last decades, research has been focused on finding biomarkers that aid in predicting diagnosis and prognosis. Although many molecules have been proposed, only a few are used in practice today. Biomarkers are objective indicators that characterize normal to pathogenic processes or pharmacological responses to therapeutic intervention. An ideal biomarker must be highly sensitive and specific to the provided data, reproducible, and cost-effective [13].

The scope of this review is to examine certain interleukins and molecules that show promising diagnostic or prognostic values related to sepsis-specific hepatic or renal injury, but whose clinical utility is still under investigation. We have excluded biomarkers that have been extensively reviewed in recent years, such as IL-6, interferons (IFNs), neutrophil gelatinase-associated lipocalin (NGAL), kidney injury molecule 1 (KIM-1), and presepsin.

## 2. Sepsis Phenotypes and Endotypes

Given that sepsis is a highly heterogeneous syndrome, the concepts of sepsis phenotypes and endotypes have emerged. Various phenotype categories divide sepsis based on temperature, hemodynamics, and organ dysfunction. Temperature profiling identified three different groups: hypothermic, normothermic, and hyperthermic, with the highest mortality among those with a greater deviation from normal temperature, either below or above. As a distinction, patients in the hypothermic group exhibit an inclination towards a tempered anti-inflammatory response compared to febrile patients, who exhibit higher plasma concentrations of both pro- and anti-inflammatory cytokines. (IL-6, TNF-α, granulocyte colony-stimulating factor, IL-10, IL-27). Hemodynamic profiling distinguishes the highest mortality among patients with a mean systolic blood pressure (SBP) around 80 mmHg. For organ dysfunction, multiple research groups have identified distinct phenotypes, with the highest mortality among patients with cardiogenic dysfunction/shock associated with either renal impairment or altered mental status [14,15,16]. One study on a substantially large cohort categorized patients into: “α phenotype” with less organ dysfunction and lower median values compared to the entire study group for IL-10, IL-6, IL-8, procalcitonin, and TNF-α; “β phenotype” with renal injury and greater comorbidities; “γ phenotype” with associated cardiovascular, hematologic, and inflammatory abnormalities, and “δ phenotype”, linked to cardiovascular and hepatic injury and which associated the highest values for IL-10, IL-6, IL-8, procalcitonin, D-dimer, and plasminogen activator inhibitor 1 (PAI-1) [17].

Concerning endotypes, multiple research groups have defined various categories, without any scholarly consensus. One proposed classification denotes the following endotypes: neutrophilic-suppressive (upregulation of innate system pathways); inflammatory (increased inflammatory response); innate host defense (moderate regulation of interleukin signaling); interferon (high expression of interferons); and adaptive (upregulation of adaptive immune pathways) [18]. Another classification indicates two extremes: macrophage activation-like syndrome (MALS) and immuno-paralysis. In MALS, a high concentration of pro-inflammatory cytokines leads to liver dysfunction, disseminated intravascular coagulation (DIC), and pancytopenia. Ferritin levels exceed 4420 nanograms/milliliter (ng/mL) and are associated with increased concentrations of IL-6, IL-18, and soluble cluster of differentiation 163 (sCD163), and a decreased ratio of IL-10/TNF-α. Immuno-paralysis is defined by a low percentage of monocytes that express human leucocyte antigen (HLA)-DR and therefore present a weak ability to initiate adaptive immune responses. Patients who did not meet either criterion were considered to fall between the two groups [19,20].

## 3. The Liver and Kidney in Sepsis

### 3.1. Acute Kidney Injury

Acute kidney injury is increasingly frequent in patients with sepsis or septic shock, contributing to worse outcomes and longer hospitalization [21]. AKI is defined by KDIGO criteria, which rely on creatinine or urine output, and in the absence of baseline information, clinicians use eGFR [9]. Crucial roles of the kidneys are maintaining fluid and electrolyte balance and filtering waste, such as exogenous drugs. Administration of intravenous fluids and antibiotics can contribute to adverse renal outcomes due to fluid overload and toxicity [22]. Sepsis-associated AKI pathogenesis (Figure 2) is not completely elucidated, but the most relevant theories are related to circulatory dysfunction, immune response, and metabolic changes. Microcirculatory dysfunction determines the loss of nurturing capillaries, microthrombi formation, and endothelial injury due to shedding of the glycocalyx and hyperpermeability. The inflammatory response in the kidneys is managed by renal tubular epithelial cells (TECs) that bind to DAMPs and PAMPs, triggering the expression of pro-inflammatory cytokines and immune cell recruitment. The complement system is mainly composed of C1–C9 components and their subunits. The activation of the complement cascade induces the formation of the membrane attack complex (C5b-C9), causing cell lysis [23,24].

### 3.2. Acute Liver Failure and Liver Dysfunction

Although the liver is less affected by sepsis, with an incidence of 11–12% sepsis-associated liver failure [12,25], its importance in pathophysiology is greater. Some of the mechanisms by which the liver protects the organism are related but not limited to: Kupffer cells that release pro-inflammatory cytokines and prevent the entrance of bacteria in the circulatory system, the recruitment and activation of neutrophils, the synthesis of complement factors, and acute-phase proteins [26]. Controlling iron metabolism is also part of liver defenses, as many microbes require iron. Sepsis-associated liver dysfunction can arise from either hypoxic hepatitis (hypoxic cell death leads to a serum increase in aminotransferases) or cholestasis (inflammation of bile ducts leads to impaired bile flow). Liver dysfunction in sepsis can progress to acute liver failure [8,26].

Another way the liver acts as a guardian is by producing clotting factors to help limit pathogens in a fibrin network. Sepsis-induced coagulopathy (SIC) represents the disruption in the fine balance of the coagulation pathway, leading to hypercoagulability in the early stages, followed by the depletion of clotting factors. After immune cells are activated by pathogens, and cytokines are released, monocytes and endothelial cells respond by releasing tissue factor (TF). TF activates the extrinsic coagulation pathway, and as sepsis progresses, anticoagulants decrease, leading to their inability to control clot formation. Fibrinolytic activity is also impaired due to elevated levels of PAI-1, which accentuates the formation of microthrombi [26,27]. SIC is an important contributor to organ injury through microthrombi, and in later stages, through the risk of bleeding due to consumption of clotting factors. Around 29% of critically ill septic patients are diagnosed with SIC [7].

## 4. Biomarkers That Pertain to AKI or ALF

### 4.1. IL-18

Interleukin-18 is part of the interleukin-1 family, and its precursor is found in Kupffer cells and other hematopoietic and non-hematopoietic cells. As a cytokine, it plays different roles depending on the receptor or other interleukins. Combined with interleukin-12, it helps T cells and natural killer cells to produce IFN-gamma. Without IL-12 or IL-15, it can lead to the differentiation of naive T cells into T helper-2 cells. Similar to other cytokines, it increases nitric oxide (NO) synthesis and cell adhesion molecules, but one specific property is Fas ligand induction, which leads to cell apoptosis. In order to minimize the immune response, IL-18 binding protein (IL-18BP) serves as the main regulatory mechanism, reducing the plasmatic levels of IL-18 [28,29].

Animal studies concluded that IL-18 not only increases cytokine levels but also helps promote serum immunoglobulin M (IgM) levels. Research has demonstrated that mice injected with *Escherichia coli* have increased serum IgM levels after multiple IL-18 injections compared to control mice, but a single shot of IL-18 only increases IFN-gamma levels [30]. Similar results were found in a study with mice that sustained burn injuries and were injected with *Pseudomonas aeruginosa.* IL-18 injections increased the mice survival rate, and serum IgM levels were higher [31].

Regarding its clinical significance, IL-18 has been shown to play a role in various diseases that display an inflammatory component. High concentrations of IL-18 have been found in patients with Crohn’s disease, asthma, chronic obstructive pulmonary disease, systemic juvenile idiopathic arthritis, and Still’s disease [28]. Given that sepsis and inflammatory disorders could clinically overlap, evidence suggests significant differences in IL-18 plasma levels between a small cohort of patients diagnosed with Still’s disease and septic patients, as the molecule displayed higher values in patients with active inflammatory disease [32].

The value of IL-18 as a mortality predictor in sepsis is disputed, as there are varying results from clinical and in vitro studies. In a study involving 40 sepsis and septic shock patients, the authors found that although IL-18 was higher at all study times in septic patients compared to healthy volunteers, there was no difference between survivors and non-survivors [33]. However, in another study involving 65 septic patients, the authors showed that patients with elevated IL-18 levels have about a 90% higher risk of death [34].

Concerning AKI, a study involving approximately 1400 patients from intensive care units found that IL-18 performed poorly to moderately regarding its ability to predict new AKI or progression of kidney injury. However, only 6.2% of those patients were diagnosed with sepsis [35]. Given that interleukin-18 is expressed in renal tubular cells, one meta-analysis shows that in patients with cirrhosis, it can differentiate between acute tubular necrosis and hepatorenal syndrome with high specificity and sensitivity [36]. SA-AKI has a complicated pathophysiology, and it is believed that it is characterized by reversible injury of the tubular cells and not by necrosis [37]. One study showed that IL-18, alongside KIM-1 and the renal resistive index, are good predictors for developing AKI in patients with sepsis, with IL-18 demonstrating the highest sensitivity [38]. The renal resistive index, an ultrasound-derived parameter, provides real-time perfusion data of the kidneys and can predict AKI [39]. Another retrospective study, which included patients with both bacterial and viral sepsis, concluded that IL-18 can be found in a wide range of elevated plasma concentrations. Mortality, AKI, and acute respiratory distress syndrome (ARDS) were correlated with IL-18 levels in both groups of sepsis [40]. Interleukin-18 serum levels rise after 6 h in the ICU, showing a significant difference between AKI and non-AKI septic patients. Nevertheless, the combination of IL-18 and NGAL performs better for AKI diagnosis, with a sensitivity of 77.9% and specificity of 94.6% [41]. Similar results were observed regarding IL-18 and the NLRP3 inflammasome, which were upregulated in septic patients compared to the control group and further increased in AKI patients versus non-AKI [42]. A study conducted on pregnant women, of which only 44% had a positive quick SOFA score on admission, concluded that IL-18 was not a good predictor for developing AKI [43]. Therefore, IL-18 could be a potential biomarker for predicting sepsis-associated AKI but not for other related causes.

Serum concentrations of IL-18 alongside IL-1β rise in a time-dependent manner in ALF models. After inhibiting pyroptosis proteins, survival rates greatly improved, liver damage was attenuated, and IL-18 concentration decreased significantly [44]. In patients with acute-on-chronic liver failure (ACLF) complicated with sepsis, serum IL-18 is significantly raised when compared to patients with ACLF with or without systemic inflammation. The authors of this study also noted that the five patients with signs of systemic inflammation and increasing IL-18 concentrations within 24 h of admission also developed sepsis [45]. IL-18 is a reliable predictor for the occurrence of hepato-biliary dysfunction (HBD) and DIC in septic patients, with an area under the curve of 0.78. A multiparametric approach, which also included IL-18BP, sCD163, soluble CD25 (sCD25), ferritin, IL-10, and other biomarkers, leads to an increase in the area under the curve to 0.93 [46]. The main key findings related to IL-18, as well as hepcidin and ferritin, can be found in Table 1.

### 4.2. Hepcidin

Hepcidin is a key molecule in iron homeostasis, primarily produced by the liver and in smaller quantities by the heart, kidneys, macrophages, and other cells. There are three isoforms, of which only hepcidin-25 is considered actively involved in iron metabolism, while hepcidin-22 and hepcidin-20 are regarded as products of degradation. Hepcidin binds with ferroportin to limit iron efflux, resulting in both anemia and reduced availability of iron for bacteria. Liver production of hepcidin is downregulated by increased erythropoiesis, while conditions of inflammation and infection promote its synthesis, especially through interleukin-6. Elevated serum hepcidin has been reported in various cancers, inflammatory disorders, and sepsis [47].

One study, which included a limited number of patients with septic shock, reported that median hepcidin values were elevated compared to normal values, with maximal concentration registered at the time of inclusion. Additionally, in five patients who developed secondary complications such as pleural effusion and pneumonia, a further increase in hepcidin levels was noted [48]. In a follow-up study, the authors concluded that hepcidin values were significantly higher in the sepsis group than in the non-sepsis control group. Hepcidin levels decreased in the subsequent days and did not correlate with 28-day mortality [49]. One paper reported that hepcidin has the potential to discriminate between survivors and non-survivors in cancer patients who develop sepsis [50]. Another research group examined hepcidin and ferritin in patients with septic shock, COVID-19, and those who had undergone surgery. Although plasma hepcidin levels did not differ substantially among these groups, the hepcidin-to-ferritin ratio was significantly lower in non-survivors compared to survivors, only in the septic shock cohort [51]. 

In mouse sepsis models, hepcidin exhibits a protective role against organ damage. Hepcidin knock-out animals displayed greater mortality rates and more severe organ damage. Hepatic injury was assessed by area of necrosis and transaminase levels, which were significantly higher in knock-out mice than in controls [52]. The protective role of hepcidin is further supported by findings in another preclinical study. The authors showed that hepcidin pretreatment in mice polymicrobial sepsis models is associated with better preserved renal function and reduced tubular injury [53]. One retrospective study involving 807 participants with severe AKI (of which 64% were diagnosed with sepsis) highlighted that low plasma concentrations of hepcidin are correlated with 60-day mortality. Further research is needed to establish if hepcidin could provide therapeutic benefits for critically ill patients [54].

Data from one study reveal that hepcidin is an independent predictor of persistent AKI in septic patients [55]. The same authors showed that serum hepcidin can be a good predictor of AKI occurrence within 7 days of admission for septic patients. Combining hepcidin with urinary NGAL leads to a sensitivity of 79.6% and a specificity of 78.3% for AKI occurrence in sepsis patients [56]. Contrasting results obtained by another research group indicated no correlation between hepcidin concentrations and the development of stage 2–3 AKI for hospitalized patients with either sepsis or non-septic conditions [57]. While hepcidin has been linked to safeguarding the organism from sepsis-associated organ damage, further clinical trials are required to establish hepcidin as a biomarker for SA-AKI.

### 4.3. Ferritin

Pertaining to iron homeostasis, ferritin is a ubiquitous protein that is pivotal in the storage of intracellular iron and can also act as an iron carrier. Serum concentrations of ferritin act as a tool to examine iron storage. In acute and chronic inflammation, serum ferritin is usually elevated; however, iron is sequestered as a defensive mechanism. Therefore, ferritin has been widely researched as a marker in various states of inflammation related to malignancies, chronic kidney disease, autoimmune disorders, and acute infections [58].

Serum ferritin has been demonstrated to have a good prognostic value in children hospitalized for sepsis or septic shock. Patients with median values between 200 and 500 ng/mL had the best outcome, while patients with concentrations exceeding 500 ng/mL had the highest mortality [59,60,61]. A large retrospective study demonstrated that for every 1000 ng/mL increase in serum ferritin values, there is a non-linear increase in mortality between 28 days and one year [62].

As a protein, ferritin is composed of heavy chains (FtH) and light chains (FtL), with its circulatory form mainly attributed to light chains. In deficient FtH mice, survival rates are significantly improved in cecal ligation puncture (CLP) models, possibly due to a compensatory increase in serum ferritin concentrations. In these mice, renal function was better preserved, and hepatic injury was limited [63]. However, light chain-deficient mice exhibit the same levels of AKI when compared to control mice; therefore, an overexpression of ferritin could protect from organ damage in sepsis, while the loss of light chains does not influence the extension of the disease [64].

Ferritin has been linked to renal recovery in patients with AKI; however, patients with sepsis-associated AKI were excluded from the study [65]. In a large retrospective cohort study, using the MIMIC-IV database, higher ferritin was associated with a longer hospital stay and higher mortality, higher SOFA, and an increased risk of AKI development. The authors noted that even in the normal range, an increase in serum ferritin is still linked with an enhanced risk of AKI compared with the low ferritin group [66]. Another study on the MIMIC-IV.2.2 database extrapolated only patients with SA-AKI. Patients with ferritin over 1056.2 ng/mL had the highest values for heart rate, temperature, creatinine, SOFA score, and in-hospital mortality [67].

Given that ferritin can be released from damaged hepatocytes, its role as a biomarker in liver injury has been investigated. Mice deficient in FtH have higher serum ferritin concentrations. Sepsis models in these mice have revealed better preservation of renal function and less damage to the liver and lungs [63]. One investigation revealed that the FtH protein could establish disease tolerance to sepsis by regulating glucose metabolism. The authors showed that both FtH-deficient mice and control mice had low blood glucose after CLP; however, the control group restored the blood glucose the following days and the FtH-deficient mice did not [68]. In clinical studies, patients with sepsis-induced liver injury displayed significantly higher levels of serum ferritin, iron, and total-iron binding capacity compared to septic patients without hepatic injury [69].

More recently, levels of ferritin have been associated with mortality in COVID-19 patients, with median values almost three times higher for non-survivors compared to survivors [70,71]. Although sepsis was present in all non-survivor cases, ferritin was not directly correlated with this outcome [70]. Ferritin can also be used as a diagnostic tool for hepatic injury in COVID-19 patients, and it has been correlated with viral clearance and hospital stay but not antibiotic use [72,73].

### 4.4. IL-27

Interleukin-27 is a cytokine with a complex role in the immune system, expressed primarily by macrophages, monocytes, dendritic cells, and other endothelial or epithelial cells. Through its receptor, it was initially believed to have a pro-inflammatory role, as it induces interferon-gamma production; however, it has been proven to also produce an anti-inflammatory response [74]. Studies on mice have demonstrated that blocking the IL-27 receptor (IL-27R) leads to a decrease in mortality in sepsis models [75]. Alveolar macrophage killing capacity is also higher in mice lacking IL-27R, as the inflammatory response in these mice offers better protection against *Pseudomonas aeruginosa* infection [76]. For human studies, a meta-analysis showed that IL-27 is a good diagnostic biomarker for sepsis both in pediatric and adult settings. With a sensitivity of 0.84 and a specificity of 0.71, it has similar accuracy to procalcitonin and presepsin [77].

Serum concentration of IL-27 has also been shown to be higher in patients with sepsis-related acute hepatic injury compared to septic patients without liver injury. The increase in IL-27 was not associated with ICU stay [78]. The same authors confirmed that IL-27R-deficient mice had less severe histological hepatic damage, therefore confirming its role as a pro-inflammatory cytokine in the process of sepsis-induced ALF [78]. Further research indicates that in mouse models with CLP-induced infection, IL-27 exacerbates liver injury. Both serum levels and hepatic expression of IL-27 are increased after LPS stimulation. Gadolinium chloride (which inhibits Kupffer cells) pretreatment can lead to reduced concentrations of IL-27, attenuating liver damage but not injury to other organs, such as the lungs and kidneys [79]. Similar results were obtained in a follow-up study, with IL-27 knock-out mice presenting fewer inflammatory factors and attenuated hepatic injury. The authors determined that IL-27 promotes macrophage pyroptosis, leading to aggravated ALF [80].

Mouse models with induced AKI demonstrate that IL-27 treatment can minimize kidney damage, expressed by less tubular injury, increased production of anti-inflammatory cytokines, and decreased production of TNF-alpha, IL-6, and IL-17A [81]. In the previous study, AKI was induced by ischemia reperfusion, which can be a pathophysiological mechanism by which sepsis causes AKI [22]. Serum concentrations are also higher in sepsis-associated myocardial dysfunction compared to the control group. However, mouse studies showed that for the myocardium, IL-27 plays an anti-inflammatory role, as pretreatment with IL-27 decreases myocardial injury biomarkers [82]. The main key findings related to IL-27, as well as the following biomarkers: IL-17A, IL-10, IL-33/ST2, high-mobility group box-1, proenkephalin (PENK), C-X-C Motif Chemokine Ligand 9 (CXCL9), soluble CD163, soluble CD25, can be found in Table 2.

### 4.5. IL-17A

The Interleukin-17 family is a group of molecules that play a role in the inflammatory response against infection. They have the ability to stimulate macrophages, dendritic cells, and others to produce chemokines, cytokines, and matrix metalloproteinases. The most common protein from this group is IL-17A, mainly produced by a subset of CD4 T cells called Th17 cells [83].

Considering the newer hypothesis regarding SA-AKI pathogenesis, IL-17A has been studied in septic mice to observe its role in renal function. Data show that IL-17A knock-out mice have a better survival rate than wild-type mice under sepsis conditions. Histological changes pertain to ameliorated tissue damage and a lower tubular injury score [84]. Therefore, IL-17A could become a novel target for managing SA-AKI therapy. Another study on mice divided into a multiple organ dysfunction (MODS) group, LPS-challenged group, and sham-operated group, indicated that IL-17 levels decreased in the MODS group, gradually over time, possibly being consumed by the kidney. The same study included patients hospitalized with sepsis, and results expressed that IL-17 levels are higher in SA-AKI compared to sepsis associated with chronic renal failure [85].

One paper highlights the importance of the Th17 ratio to regulatory T cells in septic patients with and without AKI. The authors found that the Th17/Treg ratio could be used as a predictive factor for SA-AKI development and that IL-17 concentrations were higher in the group with acute kidney injury [86]. Similar results were obtained by another research team, where IL-17A levels were proportionately higher as the AKI stage was worse [87].

In patients with ACLF complicated with sepsis, IL-17 serum concentrations are highest when compared to patients with ACLF or liver cirrhosis. There is a significant difference between patients with ACLF and patients with cirrhosis, which could indicate that IL-17 plays a role in acute liver injury [88].

### 4.6. IL-10

The interleukin-10 molecule is part of the IL-10 superfamily, mainly produced by monocytes, macrophages, and T cells, with other immune cells capable of synthesizing this cytokine. Although many cytokines have pro-inflammatory roles, IL-10 is one of the most potent anti-inflammatory and immunosuppressive molecules. It can decrease or inhibit antigen presentation, phagocytosis, the expression of major histocompatibility complex (MHC) class II, and the production of IL-1 or TNF-alpha [89]. IL-10 can suppress IL-6 secretion from Kupffer cells when the hepatic cells are stimulated with endotoxin [90]. IL-10 plays a substantial role in the cytokine storm during sepsis. IL-10-deficient mice exhibit prolonged signs of illness and higher mortality after undergoing CLP. In septic patients, NK cells increase their IL-10 production [91].

As mentioned in Chapter 4.1, IL-10 levels are elevated in patients with liver impairment (Figure 3). In the subset of septic patients with HBP and DIC, IL-10 correlated with ferritin and IL-18, in contrast, with septic patients without HBD and DIC [46]. In the previously noted study regarding patients with ACLF and sepsis, the serum concentration of IL-10 was significantly increased compared to patients with ACLF or liver cirrhosis [88]. In pediatric septic patients, IL-10 could be used to differentiate children with organ dysfunction from those with a hyperinflammatory state. The sensitivity of diagnosis for IL-10 was 89.6% for concentrations >18.4 picograms (pg)/mL. The most common types of organ dysfunction in this study were low platelets and high bilirubin, indicating liver impairment [92].

In mice, IL-10 plays a crucial role in the development of the anti-inflammatory response in septic AKI. In the first hours after inducing sepsis conditions, there is an increase in pro-inflammatory response, and at 24 h, there is an increase in IL-10. For ischemic AKI, cytokine levels were lower, and there was no peak in IL-10 levels [93]. Findings highlight that IL-10 levels are higher in patients with septic AKI compared to sepsis patients without AKI, and the receiver operating characteristic curve (ROC) analysis for predicting AKI was nearly as good as NGAL [94].

### 4.7. IL-33 and ST2

Interleukin-33 is part of the IL-1 superfamily, and it binds to Interleukin 1 receptor-like 1 (ST2), also part of the IL-1 receptor superfamily. IL-33 is produced mainly by endothelial and epithelial cells, stromal cells such as fibroblasts, and smooth muscle cells. When tissue damage happens, IL-33 is released into the extracellular space and acts as an alarmin, alerting immune cells [95,96]. ST2 has been especially studied as a marker for myocardial fibrosis with good predictive value for mortality in heart failure patients. However, it could also play a role in distinguishing the presence or absence of sepsis [97,98].

In septic mice, IL-33 treatment reduces mortality compared to control group mice. In the same experiment, pretreated mice showed a marked increase in neutrophil migration to the infection site, which led to an improvement in bacterial clearance [99]. IL-33 also promotes NK cells’ activation in septic mice, and IL-33 treatment has a beneficial role in the early stages of sepsis but not in the late phases [100].

One study aimed to investigate the differences in the cytokine profile between septic patients with and without metabolic liver disease. Patients with liver disease had higher concentrations of IL-17 and IL-33 both on day 1 and day 5 of hospitalization. Patients in the metabolic disease group also had a significant increase in serum IL-33 between admission and day 5 and a decrease in IL-10. In the same category, IL-33 had a negative correlation with sepsis severity [101].

IL-33 plays a role in AKI, as its inhibition through ST2 can provide protection against functional and histological changes in mice induced with cisplatin [102]. In a hospital setting, patients with AKI who survived showed lower serum ST2 concentrations; therefore, the IL-33 receptor could act as a prognostic marker. Although the most common cause of AKI was sepsis, ST-2 could not distinguish between the different etiologies of AKI [103].

### 4.8. HMGB1

High-mobility group box-1 is part of a group of chromosomal proteins that play an essential role in sustaining life. As a nuclear protein, it binds deoxyribonucleic acid (DNA) and enhances transcription, replication, and repair. However, when released in the extracellular space, it acts as a potent pro-inflammatory mediator [104]. Due to its dual intra- and extracellular activity, HMGB1 has been studied in relation to different types of cancers, correlating to poor prognosis, and to sepsis. In murine polymicrobial sepsis models, HMGB1 levels usually peak late, associating with death [105]. Early clinical studies concluded that in sepsis patients, HMGB1 also peaks late and associates with the severity of organ dysfunction, but it cannot distinguish between survivors and non-survivors [106,107]. Nevertheless, academics revealed that late peaks of HMGB1 associated with inflammatory disease in infectious patients act as a strong risk factor for mortality [108].

HMGB1 sustains a pro-inflammatory loop in mice with induced acute liver failure by LPS and D-galactosamine hydrochloride (D-gal). When antibodies were used to neutralize HMGB1, levels of alanine transaminase were back to normal, and liver histology was similar to the control group [109]. HMGB1 accentuates liver injury by pyroptosis of macrophages in murine CLP models [110]. Comparable results were obtained by another group of authors on ALF and AKI animal models. After inhibiting the TNF-alpha pathway, serum concentrations of HMGB1, IL-1, and IL-18 decreased, and liver and renal function were drastically improved [111].

In CLP models, HMGB1 can also localize in renal tubular epithelial cells and pass into urine, unlike in healthy settings. In these conditions, HMGB1 can interact with TECs and promote active secretion of IL-1 and IL-6, which can aggravate sepsis and SA-AKI [112].

### 4.9. Proenkephalin

Proenkephalin and other biologically active enkephalins are a result of preproenkephalin A cleavage. This family of molecules act as endogenous opioids. PENK has gained interest due to the fact that the second highest density of opioid receptors is found in the kidney. Depending on the type of receptor, enkephalins can influence diuresis and natriuresis, and reduce kidney inflammation. PENK has been mostly studied in relation to AKI and septic AKI, as it can be freely filtered through the renal glomerulus [113,114].

In one study (preprint) that included 529 patients with sepsis or septic shock, AKI was diagnosed in 44% of subjects. PENK levels on admission were correlated with AKI severity, resulting in 72% sensitivity and 83% specificity to diagnose AKI in the first 48 h [115]. In the ALBIOS trial, 28% of patients developed AKI within 48 h, from a total of 895 patients. PENK concentrations were notably elevated in patients with AKI than in those without. Among those with AKI, PENK was lower in patients who improved renal function than in those who did not; therefore, PENK could be a candidate for predicting renal function recovery [116]. One meta-analysis, which included 11 studies, concluded that PENK shows an area under the curve of 0.77 for early AKI diagnosis. From the included studies, only four comprised sepsis-associated AKI [117].

PENK also shows promising value for predicting sepsis severity and mortality, as concentrations were significantly higher in non-survivors compared to survivors [118,119].

### 4.10. C-X-C Motif Chemokine Ligand 9

Chemokines are key drivers for chemotaxis, differentiation, and multiplication of leukocytes. C-X-C motif chemokine ligand 9, alongside CXCL10 and -11, determines the recruitment of cytotoxic lymphocytes, natural killers, and macrophages. Immune cells release CXCL9 in response to IFNγ [120]. One group of authors proposed IFN-driven sepsis (IDS) as a new sepsis endotype, besides immuno-paralysis and MALS. They demonstrated that almost 20% of sepsis patients, from a cohort of 5503, display IDS characteristics. As IFNγ is the sole trigger for CXCL9, this marker had higher values in the IDS category compared to MALS. Although the greatest mortality rate was registered in the MALS category, IDS is independently associated with 28-day mortality. Serum ferritin and IL-18 displayed the highest concentration in the MALS group, followed by the IDS and immuno-paralysis groups [121].

CXCL9 has been proven to distinguish between acute interstitial nephritis and other causes of AKI, but none were related to sepsis. It has also been associated with tubular inflammation, and low urinary levels can exclude kidney injury related to infection [122,123]. To the best of our knowledge, CXCL9 has not yet been linked to SA-AKI.

C-X-C motif chemokine receptor 3 (CXCR3) is the ligand for CXCL9, and in response to IFN, hepatocytes secrete CXCR3. In ischemia reperfusion models in mice, levels of CXCL9 are increased, leading to T cell infiltration. Blocking the interaction between CXCL9 and CXCR3 increases survival, and hepatocellular damage is reduced [124].

### 4.11. Soluble CD163

CD163 acts as a receptor for the hemoglobin–haptoglobin complexes, and it is expressed on the surface of macrophages, monocytes, and lymphocytes. After pathogen recognition, monocyte activation determines an increase in sCD163, with the role of depleting iron sources for bacteria. In the later phases of sepsis, sCD163 has been shown to lead to the release of IL-10 [125]. In small clinical settings, sCD163 levels were higher in septic patients compared to controls and correlated with IL-10. sCD163 was also statistically significantly higher in non-survivors, as patients with better outcomes displayed gradually declining serum levels [126,127,128]. Another group of authors showed that sCD163 is markedly increased in septic shock, and the sCD163/IL-18 ratio in septic and septic shock patients is significantly lower than in controls. For every 1 unit increase in this ratio, there was a 3.9-fold increase in the odds of death, but the results were not statistically significant [129].

Urinary sCD163 has been proposed as a diagnostic tool for SA-AKI, with a sensitivity of 80% and a specificity of 56%. The authors correlated urine sCD163 levels with serum sCD163 and discovered good concordance [130]. sCD163 concentrations have been linked to mortality in patients with acute liver failure of different etiologies [131]. In cirrhotic patients, median sCD163 levels are significantly elevated in those with acute decompensation and infection compared to those without infection [132].

### 4.12. Soluble CD25 

The IL-2/IL-2 receptor (IL-2R) pathway plays a complex role in the immune system, establishing immune tolerance. IL-2R is composed of multiple chains, of which the α chain (CD25) is shed from T cells upon activation. Soluble CD25 has been researched as an indicator of immune-mediated disorders [133].

As a sepsis biomarker, sCD25 has been shown to display higher serum concentrations compared to controls, alongside sCD163 and IL-18; however, the author did not perform any multiparameter analysis [134].

Concerning SA-AKI, sCD25 has been investigated together with IL-10 and NGAL. Results showed that sCD25 was significantly elevated in septic AKI compared to both sepsis without AKI and AKI of other causes. In this context, sCD25 performed closely to NGAL regarding SA-AKI diagnosis, and their combination had the highest performance [94].

## 5. Future Directions and Study Limitations

Future directions regarding the integration of biomarkers in sepsis classification, diagnosis, and prognosis relate to the use of machine learning, a multiparametric approach, and personalized therapy. Machine learning utilizes large data sets to generate prediction algorithms and uncover hidden patterns. Data may take the form of biomarkers, metabolomics, and gene expression, which help create sepsis profiles, assess mortality prognosis, and support treatment decisions [135]. Specificity and sensitivity for sepsis-related organ injury can be significantly enhanced by using multi-biomarker panels, as they represent various aspects of the immune response. However, this approach faces limitations due to its cost, availability, and lack of standardization [136]. In addition to diagnosis and prognosis, biomarkers may also be used for personalized therapy in the form of agonists or antagonists. For example, hepcidin agonists could maintain renal function in septic conditions, and using antibodies against HMGB1 prevents ALF [53,109].

This review has several limitations. First, we limited our search to articles published in the English language, which may have led to the exclusion of pertinent studies in other languages. Second, the studies included in our review displayed considerable heterogeneity regarding study design and patient demographics, limiting our ability to draw uniform conclusions. Third, much of the presented evidence is derived from small-scale studies, which hinders the predictive validity of biomarkers. Fourth, we found limited data evaluating multiparametric approaches, which could improve diagnostic accuracy.

## 6. Conclusions

Sepsis is not only a heavy burden on the individual but on the medical society as well. While every organ can be gravely impacted by sepsis, acute kidney injury remains the most frequent organ dysfunction, increasing mortality risk, and liver damage can vastly influence the inflammatory response and modulate injury in other organs. While consecrated molecules such as C-reactive protein, procalcitonin, and presepsin perform well in practice, there is still a need for more specific diagnostic and predictive biomarkers for sepsis and related organ dysfunction. The biomarkers described in this review have multiple functions and could potentially be used in either AKI or ALF settings. IL-18, hepcidin, PENK, and sCD25 are good candidates for SA-AKI diagnosis. The sensitivity and specificity of IL-18 and hepcidin are enhanced by combining with NGAL. PENK also displays prognostic value for renal recovery, and ST-2 and IL-17 could be predictive of AKI severity and mortality. Liver injury is assessed by ferritin, IL-27, HMGB1, and CXCL9, mainly in preclinical studies, which leaves a gap in the knowledge regarding their abilities to act as biomarkers for SALI in clinical environments. We consider that biomarkers hold great potential for characterizing the complexity of sepsis syndrome, as various phenotypes display different biomarker serum levels and associate with different types of organ dysfunction. Further research is needed to establish the utility and practicality of novel biomarkers in clinical settings, with the goal of enhancing diagnostic accuracy and improving therapeutic choices.

## Figures and Tables

**Figure 1 ijms-26-04825-f001:**
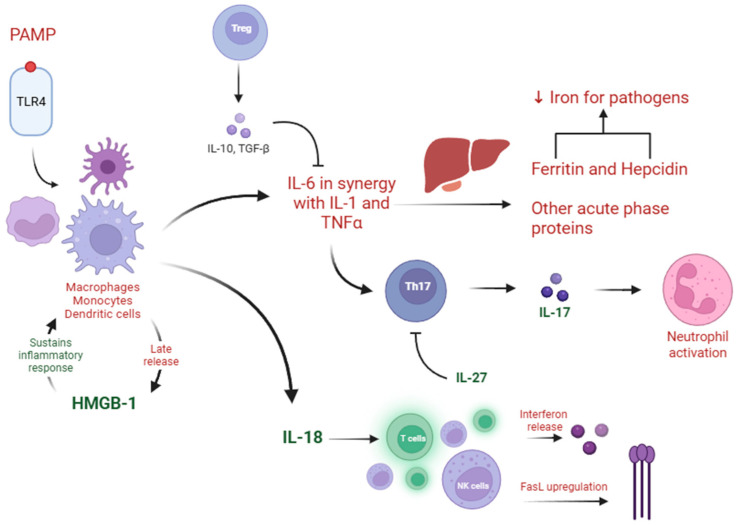
Overview of sepsis pathophysiology. FasL: Fas Ligand; HMGB1: high-mobility group box-1; NK: natural killer; Th: T helper cells; TLR: Toll-like Receptor; Treg: Regulatory T cell. Created in BioRender. Pasare, A. (2025) https://BioRender.com/0fgjo31. Accessed on 1 May 2025.

**Figure 2 ijms-26-04825-f002:**
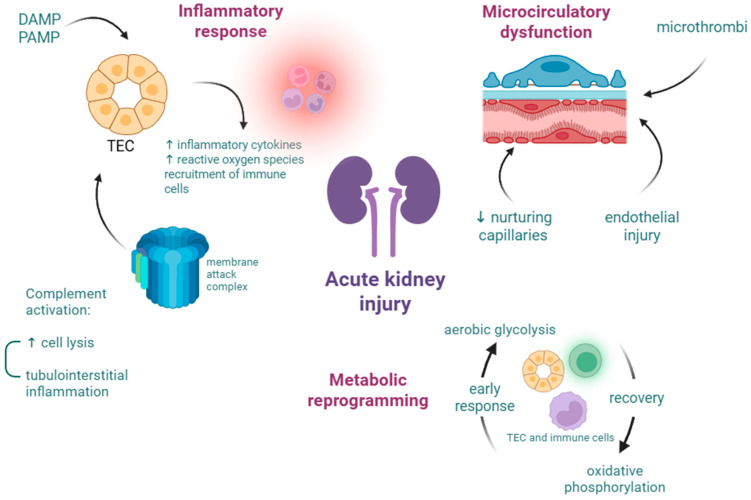
Overview of SA-AKI pathophysiology. Created in BioRender. Pasare, A. (2025) https://BioRender.com/1f9k56c. Accessed on 11 May 2025.

**Figure 3 ijms-26-04825-f003:**
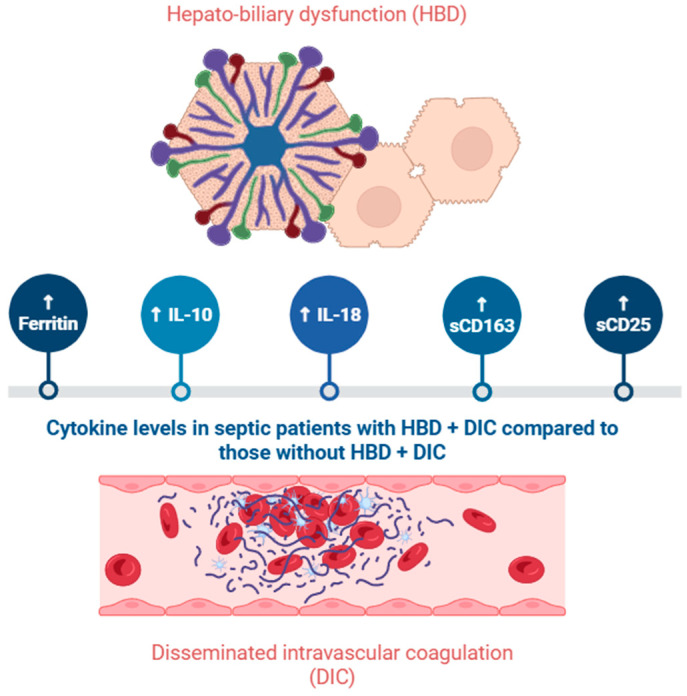
Levels of biomarkers in septic patients with and without liver impairment. Created in BioRender. Pasare, A. (2025) https://BioRender.com/g7vqax2. Accessed on 11 May 2025.

**Table 1 ijms-26-04825-t001:** Summary of advantages and limitations for IL-18, hepcidin, and ferritin.

	Advantages	Limitations
IL-18	- Higher concentrations in septic patients - Correlated with sepsis mortality - Good predictive value for SA-AKI in multiple studies - High concentration in HBD + DIC	- Elevated in other inflammatory conditions - Studies with small study population
Hepcidin	- Levels correlate with sepsis diagnosis - Hepcidin treatment lowers mortality rates (preclinical studies) - Correlated with mortality for AKI patients	- Elevated in hereditary hemochromatosis and some cancers - Predictive value for SA-AKI is improved by combining with other biomarkers - Mortality for AKI patients is not specific for SA-AKI.
Ferritin	- Good prognostic value for adult and pediatric sepsis - Serum values linked with AKI risk in septic patients - FtH protects mice from hepatic injury in sepsis models - High concentration in HBD + DIC	- Not specific for sepsis inflammation - Few clinical studies related to sepsis associated with liver injury

**Table 2 ijms-26-04825-t002:** Summary of advantages and limitations for IL-27, IL-17A, IL-10, IL-33/ST2, HMGB1, Proenkephalin, CXCL9, sCD163, sCD25.

	Advantages	Limitations
IL-27	- Similar diagnostic capacity as procalcitonin - Correlated with sepsis-associated liver injury in both preclinical and clinical studies	- Not specific for liver, investigated for sepsis myocardial dysfunction
IL-17A	- Targeted therapy leads to reduced renal injury in septic conditions	- Not sufficient clinical studies to sustain preclinical data
IL-10	- Associated with hepatic dysfunction in septic patients	- Not sufficiently specific - Reflects immune response and not direct injury
IL-33 and ST2	- Linked with metabolic liver disease in septic patients - Correlated with mortality in AKI patients	- Mostly studied in relation to heart failure - Not specific for SA-AKI
HMGB1	- Targeted therapy leads to reduced renal and hepatic injury in mice sepsis models	- Not validated in large clinical trials
Proenkephalin	- Good predictive value for SA-AKI severity and function recovery	- Not yet established to provide better diagnosis over validated biomarkers
CXCL9	- Potential to define a new sepsis endotype	- Not enough data for organ-specific injury
sCD163	- Urinary concentrations display diagnostic capacities for SA-AKI	- Not yet validated in large clinical trials
sCD25	- Similar diagnosis ability as NGAL for SA-AKI	- Further validation needed

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
