# Peer review of "Biomarkers as Beacons: Illuminating Sepsis-Associated Hepato-Renal Injury"

_ijms, 2025, doi:10.3390/ijms26104825_

Round 1
Reviewer 1 Report
Comments and Suggestions for Authors
General comment.
This is an interesting review article that discussed selected biomarkers in the diagnosis, management and risk assessment pf patients with sepsis with particular emphasis on renal and hepatic injury. The latter with limited high index biomarkers.
Although conflicting evince for the selected biomarkers were presented, the article requires major revision in terms of the relevant information and relevant roles of each of the selected biomarker specifically to sepsis.
The review article will benefit from extensive revision for ease of read and follow up of argument. Although several relevant and important points were mentioned for each of the biomarkers, some were sporadic, irrelevant and felt out of synch.
Specific items:
The statistical data on sepsis prevalence and mortality is outdated to 2015. Updated data would be helpful.
Citations mentioned in text are inconsistent often just names and sometimes names and numbers (the earlier makes it difficult to find / search).
SA-AKI (? Sepsis-Associated or Induced-Acute Kidney Injury)
Although, IL-18 was described as potential makers, its value remains in question. This was the same for a number of the biomarkers reviewed. A table summary of advantages / limitations for each of the biomarkers may be helpful.
Similarly, hepcidin was mentioned and proposed as biomarker of AKI related to spies, several limitations and non-specific findings preclude its value in evaluation of the degree of sepsis.
Line 206 with reference to ferritin is confusing. What is meant by FTH?.
What is CLP and FTH (line 206). ? ACLF. The authors introduce abbreviation without description which is confusing.
It is not clear as to why the authors limited their review to the selected few biomarkers.? (line 48-49).
Although some of the selected markers exhibited markedly sensitivity for inflammation / infection, they appear to lack specificity towards sepsis induced morbidity.
Liver enzymes mentioned without levels
IL-18 binding protein was mentioned as a specific regulatory protein for its activity?. This need to be clarified / re-warded to describe whether this is primary or secondary regulatory mechanism.
What is meant by renal resistive index (line 115)
Comments on the Quality of English Language
Some corrections would make it easy to read / follow the argument.
Reviewer 2 Report
Comments and Suggestions for Authors
The paper by Pasare et al is indeed well-written but imperfect.
The authors do not state why they selected the specific cytokines and other markers they selected. Key inflammatory cytokines as Interferon-gamma (or preferably CXCL9), TNfa and Interleukin-6 are not discussed? Key markers of T-cell activation as eg soluble-Interleukin-2-receptor (sCD25) or putative of macrophage function as eg soluble CD163 are not discussed.
I do not find sufficient focus on interplay between different markers and the given markers relation to specific sepsis phenotypes.
There is no focus on complement mediated kidney dysfunction and liver mediated coagulopathy - phenomena that have major impact on many patients with severe sepsis.
I miss a final conclusion from the authors.
Some references eg ref 8 and ref 52 are in rather obscure journals without major relevance to the topic of the papers..
Yours
Reviewer 3 Report
Comments and Suggestions for Authors
The paper reviews some new molecules as biomarkers potentially useful in the diagnostic workup of sepsis. Interestingly Authors refer to kidney and liver damage induced by sepsis.
The text is a series of descriptive parts that can be a bit boring. Is it possible to insert some figures that illustrate the physiopathological mechanisms underlying the application of the different biomarkers? Or even some summary tables?
The narrative and speculative nature of the review could also be improved through the inclusion of possible relationships between the reported biomarkers, also in the perspective of a multiparametric or score-based approach.
Round 2
Reviewer 1 Report
Comments and Suggestions for Authors
This revised version is much improved. Howevere, there remain much room for improvement:
1) Figure 1 abbreviation need definition e.g. FasL ?
2) Line 123 what is meant by "the membrane attack complex"?
3) Line 138-139 which liver enzymes? Are liver enzymes markers of liver failure of function or of liver injury?
4) Line 142 ? sepsis associated or sepsis induced (coagulopathy)?
5) Line 223 what is meant by "researchers have also "shone ""?
6) L223-262 section on hepcidin is confusing and will need to be revised for clarity and ease of follow up/read.
Comments on the Quality of English Language
This revised version is much improved. Howevere, there remain much room for improvement:
1) Figure 1 abbreviation need definition e.g. FasL ?
2) Line 123 what is meant by "the membrane attack complex"?
3) Line 138-139 which liver enzymes? Are liver enzymes markers of liver failure of function or of liver injury?
4) Line 142 ? sepsis associated or sepsis induced (coagulopathy)?
5) Line 223 what is meant by "researchers have also "shone ""?
6) L223-262 section on hepcidin is confusing and will need to be revised for clarity and ease of follow up/read.
